# MET Amplification as a Resistance Driver to TKI Therapies in Lung Cancer: Clinical Challenges and Opportunities

**DOI:** 10.3390/cancers15030612

**Published:** 2023-01-18

**Authors:** Kang Qin, Lingzhi Hong, Jianjun Zhang, Xiuning Le

**Affiliations:** 1Department of Thoracic/Head and Neck Medical Oncology, The University of Texas MD Anderson Cancer Center, Houston, TX 77030, USA; 2Department of Imaging Physics, The University of Texas MD Anderson Cancer Center, Houston, TX 77030, USA

**Keywords:** NSCLC, MET amplification, TKIs, resistance mechanism, detection, diagnosis, combined-therapy

## Abstract

**Simple Summary:**

Secondary amplifications/copy number changes of the gene *MET* (*MET* protocol oncogene) play a significant role in the development of resistance to targeted drugs in advanced non-small cell lung cancer (NSCLC). In this review, we aim to clarify the biological mechanisms of *MET* amplification-mediated resistance to tyrosine kinase inhibitors, discuss the challenges of commonly used assays for the identification of *MET* amplifications. We also summarize the latest findings on combined strategies to overcome acquired *MET* amplification-mediated resistance, especially the combinatory regimens with EGFR-TKIs and MET-TKIs.

**Abstract:**

Targeted therapy has emerged as an important pillar for the standard of care in oncogene-driven non-small cell lung cancer (NSCLC), which significantly improved outcomes of patients whose tumors harbor oncogenic driver mutations. However, tumors eventually develop resistance to targeted drugs, and mechanisms of resistance can be diverse. *MET* amplification has been proven to be a driver of resistance to tyrosine kinase inhibitor (TKI)-treated advanced NSCLC with its activation of *EGFR*, *ALK*, *RET*, and *ROS-1* alterations. The combined therapy of MET-TKIs and EGFR-TKIs has shown outstanding clinical efficacy in *EGFR*-mutated NSCLC with secondary *MET* amplification-mediated resistance in a series of clinical trials. In this review, we aimed to clarify the underlying mechanisms of *MET* amplification-mediated resistance to tyrosine kinase inhibitors, discuss the ways and challenges in the detection and diagnosis of *MET* amplifications in patients with metastatic NSCLC, and summarize the recently published clinical data as well as ongoing trials of new combination strategies to overcome *MET* amplification-mediated TKI resistance.

## 1. Introduction

Non-small cell lung cancer (NSCLC) constitutes around 85% of lung cancer, which has been the leading cause of cancer-related deaths worldwide [1]. In the past decade, groundbreaking progress in personalized therapy and targeted agents has led to unprecedented clinical improvements in the subgroup of patients with advanced or metastatic NSCLC carrying active oncogenic driver alterations. Molecular profiling to identify actionable oncogenic drivers is now recommended as part of the initial clinical work-up for patients with metastatic NSCLC. Currently, In NSCLC, especially of the non-squamous histology, predictive biomarkers recommended for testing by the NCCN profiling panel are *EGFR*, *KRAS*, and *BRAF* mutations; *ALK*, *RET*, and *ROS1* gene rearrangements; *MET* alterations including *MET* exon 14 skipping mutations and *MET* amplifications; *ERBB2 (HER2)* mutations; and *NTRK* 1/2/3 gene fusions [2]. Although the well-established targeted drugs show outstanding efficacy in initial disease control, drug resistance always inevitably develops. Clarifying and overcoming the resistance to targeted drugs with novel strategies is one of the major challenges in the era of personalized therapy.

The MET proto-oncogene (hereafter referred to as *MET*) encodes the receptor tyrosine kinase or hepatocyte growth factor (HGF) receptor, which, along with its ligand HGF (HGF/MET axis), functions as an essential regulator of cell survival, proliferation, motility and migration. Dysregulation of MET signaling has been found in a variety of cancers through different mechanisms, such as activating point mutations of the *MET* gene, overexpression of the ligand HGF, *MET* gene copy number gain (*MET*-CNG)/amplification, and *MET* gene fusions [3,4].

*MET* amplification occurs in 1–6% of NSCLC cases and was considered as a negative prognostic factor [5,6,7]. In recent years, increasing evidence has implicated that *MET* amplification was a key driver of acquired resistance to these aforementioned targeted therapies such as EGFR-TKIs and ALK-TKIs. Although an increasing number of drugs acting on MET signaling is currently achievable, for example, the MET/ALK/ROS tyrosine-kinase inhibitor crizotinib or selective MET-TKIs (capmatinib, savolitinib, tepotinib) [7], more strategies are needed to overcome *MET* amplification-mediated acquired resistance to TKIs. Therefore, it is important and necessary to clarify the underlying molecular mechanisms of *MET* amplification-mediated resistance, and to find out appropriate ways to identify *MET* copy number gains and amplifications, so that researchers can develop effective therapeutic strategies to overcome this resistance and prolong the life of NSCLC patients. Our review focuses on the molecular mechanisms of acquired resistance to targeted therapies mediated by *MET* amplifications, and the ways and challenges in detection and diagnosis of *MET* amplifications in NSCLC. We also summarize the recently published clinical data as well as the ongoing trials focusing on new combination strategies to overcome *MET* amplification-mediated TKI resistance.

## 2. MET Biology, Structure, Function, and Pathways

The receptor tyrosine kinases (RTKs) are encoded by a family of proto-oncogenes with more than 75 members that regulate cellular growth, oncogenesis, tumor metastasis, and progression through downstream signaling pathways such as the RAS-RAF-MEK-ERK and PI3K-AKT-mTOR pathways [8]. *MET*, together with *EGFR*, *ALK*, *BRAF*, etc. are all members of this family, which were found to be frequently mutated in advanced NSCLC [9]. Tyrosine kinase inhibitors bind and act on these RTKs, and lead to the inhibition of downstream signaling pathways which would otherwise induce tumor cell growth and proliferation [10]. In patients with advanced NSCLC undergoing TKI treatments, acquired resistance always develops and limits the long-term application of these targeted agents. Bypassing the activation of *MET*-related pathways has proven to be one of the underlying reasons [11].

The human *MET* gene is a 120 kb proto-oncogene that is located on chromosome 7 band 7q21–q31. Hepatocyte growth factor receptor (HGFR) or MET protein is the product of the *MET* proto-oncogene, and its ligand HGF is a disulfide-linked a-b heterodimeric molecule, also known as plasminogen-related growth factor-1(PRGF-1) [12]. MET protein is normally expressed in various epithelial and mesenchymal cell types. Upon HGF binding, the HGF/MET signaling pathway is activated, then MET undergoes homodimerization and autophosphorylation of a series of tyrosine residues within the intracellular region, including Y1230, Y1234, Y1235, Y1313, Y1349, and Y1356, etc., which lead to the activation of multiple intracellular signaling pathways including the RAS-RAF-MAPK, JAK-STAT and PI3K-AKT/mTOR, and phospholipase C pathways [13] (Figure 1). The signalings have been shown to trigger a variety of cellular responses, including cell proliferation, tissue regeneration, angiogenesis, and cellular invasion, etc. [14]. Oncogenic *MET* alterations, including the overexpression of MET protein or *MET* gene alterations, such as mutations, amplifications, or fusions, cause dysregulation of the HGF/MET signaling pathway, and lead to a wide range of human cancers, including papillary renal cell carcinoma, gastric cancer, and non-small cell lung cancer, etc. [3,15].

## 3. *MET* Amplification as a Mediator of Resistance to Targeted Agents in NSCLC

Increased gene copy numbers (GCN) of the *MET* gene could be observed in approximately 1–3% of NSCLC, either due to de novo amplification or as a secondary resistance mechanism in response to targeted therapies [4]. Acquired resistance to EGFR-TKIs can develop via both *EGFR*-dependent and *EGFR*-independent mechanisms. Acquisition of the Exon20 T790M mutation has been proven to be the most common *EGFR*-dependent cause, with MET signaling dysregulation as the most common *EGFR*-independent cause [16,17]. *MET* amplification-mediated resistance has a prevalence of 5–21% after first/second generation EGFR-TKI treatment, 7–15% after first-line osimertinib therapy, and 5–50% of osimertinib resistance after secondary and/or further-line osimertinib treatment [18,19,20].

The underlying mechanism by which *MET* amplification leads to EGFR-TKI resistance may be associated with phosphorylation of ErbB3 (HER3), which functions as a key activator of the PI3K/AKT and MEK/MAPK pathways, providing bypass signaling in the presence of EGFR-TKIs [17,21,22]. A study by Y.Yarden and colleagues showed that a combination of mAb33 (an anti-HER3 antibody) with cetuximab and third-generation EGFR-TKI osimertinib markedly reduced HER3, and also downregulated MET expression [21]. In another study, inhibition of MET through an inhibitor or knockdown of the *MET* gene restored the effects of osimertinib on ErbB3 inactivation and ErbB3 phosphorylation suppression [22]. Taken together, these findings suggested that phosphorylation of ErbB3 was involved in *MET* amplification-mediated acquired resistance to EGFR-TKIs in advanced NSCLC. In addition, an upregulation of mTOR and Wnt signaling proteins was observed in MET-TKIs/EGFR-TKI-resistant NSCLC cell lines, implying the role of alternative cell signaling pathways in TKI resistance [23]. Furthermore, MET-TKIs and EGFR-TKIs showed a synergistic inhibitory effect on cell proliferation and downstream activation of signal transduction. Therefore, a combination of HGF and EGF tyrosine kinase inhibitors could potentially be targeted in a synergistic fashion to overcome *MET* amplification-mediated resistance to EGFR-TKIs [24,25].

*ALK*-rearranged NSCLC is another major subtype of lung cancer, which occurs in around 3–5% of lung adenocarcinomas [26]. As a member of the RTKs family, ALK also regulates cellular proliferation and survival through pathways such as the PI3K-AKT-mTOR, RAS-RAF-MEK-ERK, and JAK-STAT pathways [27,28]. Around 50% of resistance to second-generation ALK-TKIs (ceritinib, alectinib, and brigatinib, etc.) is caused by *ALK*-independent resistance mechanisms, most often due to activation of bypass signaling pathways, including activation of MET, EGFR, and IGF-1R (insulin-like growth factor 1 receptor), etc. [29,30]. MET overactivation was shown to be involved in the development of acquired resistance to alectinib, but not to crizotinib in NSCLC cell lines [31,32,33]. MET activation-mediated resistance was found to be overcome by crizotinib, which was initially developed as a MET receptor TKI [32,33,34]. However, more evidence is needed to fully clarify the mechanism and functions of *MET* amplification in ALK downstream signaling and ALK-TKI resistance development.

*KRAS* is the most frequently mutated cancer-related driver in non-small cell lung cancer, which could be observed in over 30% of NSCLC patients. *KRAS G12C* variants are the most commonly found subtype of oncogenic *KRAS* alterations, which have been identified in around 10% of NSCLC cases [35]. Acquired focal *MET* amplification in a patient with *KRAS G12C*-mutant lung adenocarcinoma treated with sotorasib was also documented [36].

A preclinical study reported that constitutive activation of *KRAS* could lead to the persistent stimulation of downstream signaling pathways, for example, the PI3K/AKT/mTOR cascade and the overexpression of MET protein [37], which indicated that MET amplification was one of the acquired bypass mechanisms of resistance to *KRAS* inhibitors. Lito et al. demonstrated that the inhibition of SHP2, which functions as a valuable co-inhibitory target in *KRAS G12C* signaling and is also a central node in RTK and RAS inhibition signaling, was able to overcome *KRAS G12C* inhibitor resistance in vitro [38,39].

*MET* amplification is also a known resistance mechanism in *RET*-rearranged NSCLC. Data on acquired resistance to *RET*-specific inhibitors, such as selpercatinib and pralsetinib, have suggested that on-target mutations at non-gatekeeper sites or the emergence of off-target alterations such as *MET* amplification or *NTRK* fusion are potential mechanisms of acquired resistance [40,41,42]. Furthermore, combinational therapy with crizotinib, which is a MET/ALK/ROS1 TKI, with selpercatinib in patients who had *RET* fusion-positive and *MET*-amplified NSCLC showed clinical efficacy in selpercatinib-resistant tumors [40].

Although there have been many studies that investigated and identified the underlying mechanism of acquired resistance to the targeted agents mediated by *MET* amplification, this issue requires further elucidation through more preclinical and clinical studies.

## 4. Detection of *MET* Amplification and Overexpression

Since *MET* amplification is a common resistance mechanism to different TKI resistances in lung cancer and inhibitors are available to be used, it is then critical to detect *MET* amplification with appropriate methods and cut-offs so that patients can be identified to be offered potential anti-MET treatment. *MET* copy number gains can occur either as polysomy (multiple copies of chromosome 7) or true amplification (regional or focal copy number gains without chromosome 7 duplication) [43]. True amplification is more likely to lead to oncogene addiction [44]. Various assays have been developed for the detection of *MET* copy number changes. Fluorescence in situ hybridization (FISH) is the gold standard method for *MET* amplification detection. Next-generation sequencing (NGS) is becoming more popular clinically, as the results cover multiple oncogenes, and NGS profiling can be utilized for tissue or liquid biopsy/circulating tumor DNA, either DNA- or RNA-based. Immunochemistry (IHC) is mainly used for the identification of MET overexpression. Quantitative real-time polymerase chain reaction (qRT-PCR) is less commonly used. Each assay has its advantages and disadvantages.

### 4.1. Fluorescence In Situ Hybridization (FISH)

Fluorescence in situ hybridization (FISH) is the standard method, and is also the way that is mostly used clinically for identification of *MET* amplification. *MET* amplification can be defined by FISH, either by determining gene copy number or by taking the ratio of *MET* to CEP7 (centromere 7 enumeration probe). *MET* amplification is defined as *MET* GCN ≥ 5 with the Cappuzzo criteria, which means five or more copies of *MET* are detected per tumor cell [45,46,47]. Cut-off points such as a *MET* GCN of ≥6 or 10 or 15 are also used in some studies [48,49,50,51,52]. However, GCN itself cannot distinguish true amplification from polysomy. *MET* amplification can also be determined by the MET/CEP7 ratio, and a cut-off value of *MET*/CEP7 ratio round 2 is commonly used to define *MET* amplification [46,47,53,54,55,56,57].

In some studies, *MET* amplification was categorized into three degrees using the *MET*/CEP7 ratio: low amplification 1.8 ≤ *MET*/CEP7 ≤ 2.2; intermediate amplification 2.2 < *MET*/CEP7 < 5; and high amplification *MET*/CEP7 ≥ 5 [44]. Compared with GCN, the *MET*/CEP7 ratio identifies *MET* amplification more accurately when there is no concurrent chromosome 7 polysomy [58]. However, there is no consensus on a single definition cut-off value with the FISH assay; other cut-off values may also be used. For instance, in a study by Buckingham et al., tumor cells with CEN7 signals on average ≥3.6 were categorized as polysomic *MET* amplification [59].

Tumors harboring de novo *MET* amplifications (high level, i.e., *MET* to CEP7 ratio ≥ 5) are thought to be primarily dependent on the MET signaling pathway for growth, as there are often no other concurrent oncogenic drivers. These amplifications are identified in <1–5% of NSCLCs, and indicate a poor prognosis [60,61,62,63].

Furthermore, the literature suggests that, compared with other assays, FISH is unique in that it can capture the various levels of *MET* gene amplification, including “true” high-level *MET* gene amplified cases characterized by a high *MET* GCN (≥6 per cell) without concomitant polysomy (i.e., a high *MET*/CEN7 ratio) [64]. However, FISH only detects tissue samples, and it is inapplicable when tissue samples are not available, which limits its clinical use [65].

### 4.2. Next-Generation Sequencing (NGS)

Simultaneous targeted DNA- and RNA-based next-generation sequencing (NGS) offers the most straightforward and comprehensive profiling for not only *MET* amplifications, but for all treatment relevant genetic alterations, including the *MET*14-skipping aberrations, which cannot be identified by FISH but are of high clinical significance; therefore, NGS has also been widely applied in clinical practice for detection of MET copy number gains [66].

Two methods are commonly used for NGS-targeted approaches: capture hybridization-based sequencing and amplicon-based sequencing, and each has its own advantages and disadvantages. A head-to-head study compared these two types of methods, and indicated that amplicon-based approaches have a much-simplified workflow, and require smaller amounts of DNA for assessment. By contrast, hybridization-based NGS profiling was less likely to miss mutations, and performed better with respect to sequencing complexity and uniformity of coverage [67,68,69,70].

However, *MET* amplification detected via NGS is reported as continuous variables, and there is a lack of consensus on a single cut-off value. Normally, the cut-off value ranges from GCN 2.3–10. For example, in the TATTON study, *MET* amplification used a cut-off value as GCN ≥ 5 [71]; in the INC280 study [72], *MET* amplification was determined with a GCN ≥ 2.3; in the ongoing phase 2 INSIGHT 2 (NCT03940703) study, *MET* amplification was defined as GCN ≥ 6 [73].

Now that NGS is increasingly used to optimize precision oncology therapy in NSCLC, the question is whether NGS assays can replace the FISH method regarding the classification of *MET* copy number status. Copious studies have investigated this question. Heydt C. et al. [74] compared 35 *MET*-amplified NSCLC samples (including 5 samples showing a low-level *MET* amplification, 10 samples with an intermediate-level *MET* amplification, and 10 samples with a high-level *MET* amplification), and found that MET-IHC had the best agreement with MET-FISH. Furthermore, only high-level *MET*-amplified cases (GCN ≥ 6), showed better concordance between NGS and FISH detections than those in intermediate- or low-level *MET*-amplified patients. This was confirmed through a study by Schubart C. et al., which compared detection results of 205 consecutive NSCLC cases with *MET* alterations, using either an amplicon-based, 15-gene NGS panel, or the standard FISH method. Among the 205 patients detected, 9 cases were classified as *MET*-amplified by NGS, and 16 cases were classified as high-level *MET*-amplification by FISH, yielding a discrepancy of 43.7% (7/16); only cases harboring a *MET* GCN > 10 showed the best concordance when comparing FISH versus NGS (80%, 4/5) [64]. In a study by Peng et al. [75], the concordance rate among FISH and NGS was only 62.5% (25/40). In addition, amplification identified by NGS was found to be an ineffective predictive biomarker, and failed to distinguish significant clinical outcomes. The PR rate was 60.0% (6/10, with *MET* GCN ≥ 5) vs. 40.0% (12/30, with *MET* GCN < 5); the median PFS was 4.8 months vs. 2.2 months (*p* = 0.357). A study by Lai et al. also demonstrated a low concordance between the FISH assay and NGS profiling; among samples with FISH-positive results with GCN ≥ 8, only one-third were identified as *MET* amplification with NGS [76]. Of the 18/39 patients identified as MET-high (two amplifications and 16 polysomies), only 8/18 were deemed to have *MET* CNG by NGS. Of the two *MET*-amplified tumors (3.4 and 2 by ratio), the latter was reported as non-*MET*- amplified on NGS. In addition, only 1/3 tumors with a *MET* CNG greater than 8 by FISH were identified as *MET*-amplified with NGS. The result of the TATTON study also showed low consistency between NGS and FISH for *MET* amplification; among all 47 FISH-positive patients, only 12 had *MET* amplification by NGS [71]. Taken together, FISH is the standard method for the detection of various levels of *MET* amplifications during routine diagnostics; NGS is widely used, but is not yet able to replace FISH for the detection of *MET* gene copy number gains.

In recent years, the use of liquid biopsy for genomic profiling has made multi-gene sequencing more easily accessible to patients. NGS of circulating cell-free DNA (cfDNA) has also been used to detect *MET* alterations in clinical studies, including the VISION study [77] and the INSIGHT 2 study [73]. Both studies used liquid biopsy to prospectively screen patients for enrollment and establish the role of liquid biopsy as a tissue-sparing, less-invasive and more easily accessible method for the detection of *MET* alterations.

### 4.3. Immunohistochemistry (IHC)

*MET* can be transcriptionally induced in cancer cells in the setting of hypoxia/inflammation to activate proliferation, decrease apoptosis, and promote migration. Thus, tumors can rely on MET signaling, even in the absence of a genomic driver such as *MET* amplification, mutation, or fusion [78]. MET can also be overexpressed in cancers that harbor an activating genomic signature, including those with primary/secondary *MET* amplification, or *MET* exon 14 alterations. Therefore, MET protein overexpression detected by IHC is also commonly used for screening of *MET* gene amplification. Various scoring systems are currently in clinical use to define MET protein expression and overexpression. The most common way is categorizing the MET expression based on a 0–3+ scale into four degrees: negative (0), weak (1+), moderate (2+), or strong (3+). By the MetMab criteria, the cut-off for MET overexpression should be 2+ in at least 50% of the cells [79].

The H-score system multiplies the percentage of cells with 1+, 2+, or 3+ staining by the SI (staining intensity) score [80]. H-scores range from 0–300, and over 200 usually defines MET overexpression. However, cut points vary as well [81,82]. Investigators also used a median H-score (of the range of H-scores obtained from samples exclusively within a given study) as a cut point for overexpression; this approach makes standardization across studies difficult [50,83]. The H-scoring system multiplies the percentage of cells with 1+, 2+, or 3+ staining by the staining intensity score [80]. H-scores range from 0–300; ≥200 usually denotes overexpression, but cut points vary [82,84].

Whether IHC screening for MET overexpression can be used for *MET* amplification detection remains controversial, and attempts to take MET IHC as a marker of MET dependency have largely been unsuccessful [63,82,84]. MET IHC demonstrated poor correlation with the *MET*/CEP7 ratio in sarcomatoid lung cancer, regardless of the stage [82]. In a tri-institutional cohort of patients with metastatic lung adenocarcinoma, more than 30% of cases were MET IHC-positive, but only 2% were *MET*-amplified. MET IHC even failed to detect MET in two of the three *MET*-amplified patients [63]. MET IHC was not an effective predictive marker for MET-directed therapies in some clinical trials [85,86]. For example, in a study by Spigel D.R. et al. [85], the HR of PFS in patients with MET IHC 3+ status was 0.86 (95% CI, 0.58 to 1.29), compared with 1.06 (95% CI, 0.85 to 1.32) in patients with MET IHC 2+ status. Moreover, no statistically significant differences in OS, PFS, or ORR between the onartuzumab and placebo arms were observed when analyzed using MET FISH status. Coupling the reports from the growing literature, there is a strong challenge in taking MET IHC as an effective way of screening for MET dependency. Therefore, MET IHC is not viewed as an effective way of screening for MET dependency, and it is less commonly used for clinical *MET* amplification detection.

### 4.4. Reverse-Transcription Polymerase Chain Reaction—qRT-PCR

Real-time PCR(RT-PCR), also known as quantitative PCR (qPCR), is the gold-standard for sensitive, specific detection and quantification of nucleic acid targets, and is a valid method for *MET* exon 14-skipping mutation detection [87]. However, unlike gene point mutation, *MET* amplification is difficult to test with qPCR or qRT-PCR; therefore, it is less commonly used in clinical settings for *MET* amplification detection compared to FISH/NGS [16,88,89,90,91].

Although seldomly used, the detection of *MET* amplification using ddPCR shows very high concordance rates with FISH, either in tissue samples only (100%, 102/102) or among both peripheral blood and tissue samples (94.17%, 97/103). This indicates that ddPCR is an optional non-invasive method for detecting of *MET* CNG in blood samples as compared with the FISH method in tissue samples; thus, it may be an alternative method for MET amplification detection when FISH is not applicable, especially when tumor tissue is not available [65]. Again, cut-off values vary, and consensus on the standard definition for *MET* amplification by PCR remains to be proposed.

## 5. Drug Combination Strategies to Overcome Secondary *MET* Amplification Resistance

Since acquired *MET* amplification can bypass the initial oncogene driver to mediate resistance, it is reasonable to hypothesize that inhibition of MET signaling, together with continued inhibition of the initial oncogene driver, can overcome resistance. In the last a few years, there has also been much progression in the development of new agents that act on the HGF/MET pathways. MET-targeting drugs that are currently used in clinics and trials can be divided into three general categories: small molecule inhibitors (e.g., crizotinib, savolitinib, tepotinib, and foretinib), antibodies against the MET receptor (e.g., onartuzumab or amivantamab), and antibody-drug conjugates (e.g., telisotuzumab, vedotin) [92]. In some of the preclinical studies, it has been shown that adding a MET inhibitor to *MET*-amplified *EGFR*-mutant-resistant NSCLC cells can overcome resistance [93].

Therefore, numerous clinical studies have shown preliminary efficacy using this approach. In patients with *EGFR*-mutant NSCLC and *MET* amplification with disease progression on EGFR TKI treatment, subsequent treatment with an MET inhibitor and EGFR-TKI combination rendered clinical benefits in a series of phase I/II studies (Table 1) [71,72,73,94,95,96,97,98,99,100,101].

The TATTON trial [71] demonstrated the clinical benefits of osimertinib plus savolitinib in patients with previously treated *EGFR*-mutant *MET*-amplified NSCLC, with an objective response rate (ORR) of 44%. Among patients progressed on a third-generation EGFR-TKI, the ORR was 30%. Notably, an ORR of 64% was observed among 23 patients with *EGFR*-mutant T790M-negative NSCLC without prior third-generation EGFR-TKI treatments. Another study confirmed that the combination of capmatinib with geftinib demonstrated a PFS of 3.3 months, and an ORR up to 47% in patients with *EGFR* mutation and *MET* amplification (defined by CGN ≥ 6) [5]. In the INSIGHT study, the combination of tepotinib and gefitinib showed significantly a better PFS (16.6 months vs. 4.2 months) and OS (37.3 months vs. 17.9 months, respectively) than chemotherapy in patients with resistant *EGFR*-mutant NSCLC, especially in patients with high MET over-expression [96]. The ongoing ORCHARD study (NCT03944772) included 20 patients with *MET* amplification who progressed on first-line osimertininb monotherapy, and received second-line combinatory treatment with osimertinib and savolitinib [99]. Initial benefits for the patients were presented with good tolerance: among 17 patients who were evaluable for confirmed response analysis at data cut-off (DCO), 7 patients had confirmed partial response (ORR 41%, 7/17) and 7 patients had stable disease (DCR 41%, 7/17). The ORCHARD study is still ongoing, and more results are expected to be released in the future.

With those successes, pivotal trials were designed to further evaluate this combination approach to overcome *MET* amplification-medicated resistance in *EGFR*-mutant NSCLC. The data in the INSIGHT study led to the investigation of tepotinib plus osimertinib in the INSIGHT 2 trial [73]. Preliminary data from the INSIGHT 2 study suggested that the combination of tepotinib and osimertinib has activity in patients with *EGFR*-mutated advanced NSCLC with *MET* amplification who progressed on first-line osimertinib. In the first 48 patients who had over 3 months follow up, the ORR was 45.8% [95% CI, 31–61%], with duration of response not reached [73]. Similarly, results of the TATTON trial led to further development of the savolitnib plus osimertinib combination in the SAVANNAH trial [101]. SAVANNAH is a global, randomized, single-arm phase II trial that is studying the efficacy of savolitinib with osimertinib in patients with *EGFR*-mutant, locally advanced or metastatic NSCLC with MET overexpression and/or amplification, who progressed following treatment with osmiertinib. Patients were treated with savolitinib with osimertinib. Preliminary results demonstrated an ORR of 32% in the total population. In patients with high levels of MET overexpression and/or *MET* amplification (defined as IHC90+ and/or FISH10+), the ORR was high at 49% [95% CI, 39–59%] [101].

Other studies investigating the clinical evidence of dual inhibition of EGFR and MET with small molecule inhibitors or monoclonal antibodies, such as capmatinib plus geftinib, telisotuzumab plus erlotinib, savolitinib plus geftinib, onartuzumab plus erlotinib, capmatinib plus erlotinib, and emibetuzumab plus erlotinib within patients with NSCLC with EGFR-mutant and *MET* alterations, are summarized in Table 1 [71,72,73,94,95,96,97,98,99,100,101].

## 6. Conclusions

Acquired *MET* amplification functions as a mechanism of resistance to targeted therapies in NSCLC, and now has been proven to be a pharmaceutical target to overcome this resistance. Although evidence was most abundant and convincing for *MET* amplification mediated resistance to EGFR TKIs, its role in mediating resistance to other targeted therapies, such as ALK, ROS1, or RET TKIs, is being recognized. Case reports and case series have indicated that dual inhibition of MET and the initial oncogene driver pathway may be a valid clinical approach to overcome resistance. With *MET* amplification serving as a general resistance mechanism to targeted therapies in lung cancer, it is crucial to be able to detect *MET* amplification in a reliable manner, in order to identify the appropriate patients who can benefit from MET-targeting therapy.

*MET* amplification or MET overexpression could be detected by multiple clinical pathology laboratory tests, including FISH, NGS, and IHC. However, clinically meaningful cut-offs need to be standardized for continuous variables, including the copy number gain of *MET* amplification and MET overexpression. As clinical diagnostic methods migrate towards more comprehensive and technically sophisticated NGS assays, further understanding of NGS assays for the detection of *MET* amplification is needed, both in tumors and plasma, and ideally both in DNA and RNA. The effective detection of *MET*-dependent cancers is critical, given that *MET*-directed targeted therapy is active in many of these cancers. Importantly, the level of activity of *MET*-targeted therapies is associated with the degree of oncogenic addiction to *MET* pathway signaling.

To overcome *MET* amplification-mediated acquired resistance to TKIs, combination therapies to inhibit both MET and the primary driver oncogene are necessary. EGFR-MET TKI combination therapy has shown promising clinical efficacy in this setting. While awaiting the maturation of the large clinical trial results and regulatory approval of this approach, further studies are necessary that aim to standardize the *MET* amplification detection assay and the cut-off values.

## Figures and Tables

**Figure 1 cancers-15-00612-f001:**
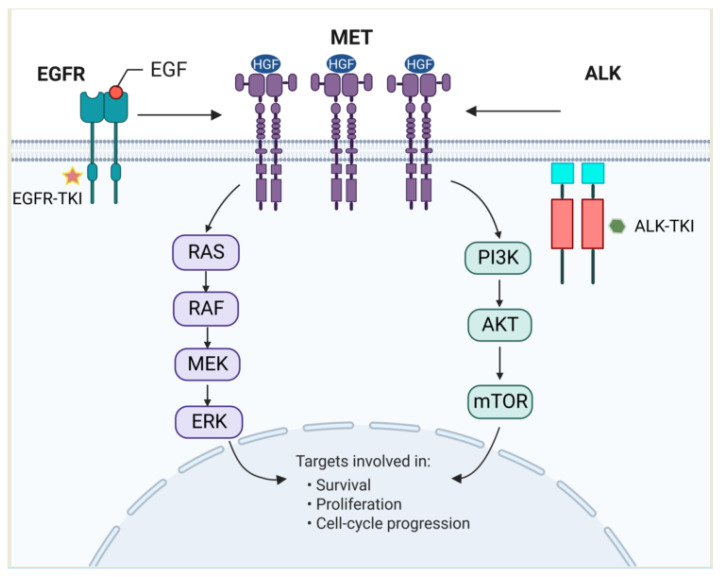
Mechanisms of *MET* amplification-mediated resistance to molecularly targeted therapies in non-small cell lung cancer (NSCLC). *EGFR* mutation or *ALK*-rearrangement as the primary driver oncogene shown. MET, EGFR, and ALK are all members of the RTK family, which regulates cellular proliferation and survival through common downstream pathways such as the PI3K-AKT-mTOR and RAS-RAF-MEK-ERK pathways.

**Table 1 cancers-15-00612-t001:** Summary of key clinical studies on combined therapies to overcome acquired MET-amplification-mediated resistance to EGFR-TKIs in EGFR-mutated NSCLC.

Study (Author, Year, NCT ID)	Treatment	Phase of Study (Number of Patients)	MET Diagnostic Assays and Criteria	Concurrent EGFR Mutations and Prior EGFR-TKIs	Lines of Therapies (Prior EGFR TKIs)	mPFS, Months	mOS, Months	ORR%
**Combined therapies with first-generation EGFR-TKIs and MET inhibitors**								
Yang et al. (2021) NCT02374645 [94]	Savolitinib plus Gefitinib	phase Ib *n* = 64 safety run-in *n* = 13 (savolitinib + gefitinib *n* = 6; savolitinib + gefitinib *n* = 7); expansion savolitinib+ gefitinib *n* = 51	MET GCN ≥5 or MET/CEP7 ratio ≥2 by FISH	EGFR-mutated advanced NSCLC	≥1 (A prior EGFR-TKI)	4.2 (95% CI: 3.5, 8.5)	NR	NR; In EGFR T790M-negative: ORR: 52% (12/23)
McCoach CE, et al. (2021) NCT01911507 [95]	Capmatinib + Erlotinib	Phase I/II *n* = 17 Cohort A (EGFR mutant *n* = 12) cohort B (EGFR wildtype, *n* = 5)	CNG or MET/CEN7 ratio outside of normal range by FISH; MET IHC 2-3+;	Cohort A: EGFR Mutant; cohort B: EGFR wildtype	≥1 prior EGFR TKI	NR	NR	Cohort A: 50%; Cohort B: 75%
Wu et al. (2020). INSIGHT study NCT01982955 [96]	Tepotinib + Gefitinib vs. Chemotherapy (pemetrexed + cisplatin or carboplatin);	Phase Ib (18)/Phase II (55)	MET OE (IHC 2+ or IHC3+) or MET amp (FISH, mean GCN ≥ 5, and/or MET/CEP7 ratio of ≥2)	EGFR-mutant, T790M-negative	≥2	Overall: 4.9 (90% CI: 3.9–6.9) vs. 4.4 (90% CI: 4.2–6.8)HR 0.67 (90% CI:0.35–1.28) In the high MET subgroup (IHC3+): mPFS: 8.3 (90% CI: 4.1–16.6) vs. 4.4 (90% CI: 4.1–6.8), HR 0.35, 90% CI: 0.17–0.74 In the MET amplification subgroup:16.6 (90% CI: 8.3–not estimable) vs. 4.2 (90% CI: 1.4–7.0); HR 0.13, 90% CI: 0.04–0.43	Phase II Overall: 17.3 (90% CI: 12.1–37.3) vs. 18.7 (90% CI:15.9–20.7); HR 0.69, (90% CI: 0.34–1.41) In the high (IHC3+) MET subgroup: 37.3 (90% CI 24·2–37·3) vs. 17·9 (12.0–20.7); HR 0.33, 90% CI: 0.14–0.76. In the MET amplification subgroup 37.3 months (90% CI not estimable) vs. 13.1 [3.25–not estimable]; HR 0.08, 90% CI: 0.01–0.51)	Phase II Overall: 45% (29.7–61.3) vs. 33% (17.8–52.1) In the high (IHC3+) MET subgroup: 68% (47.0–85.3) vs. 33% (14.2–57.7) In the MET amplification subgroup: 67% (39.1–87.7) vs. 43% (12.9–77.5)
Camidge et al. (2022) [97]	Telisotuzumab Vedotin + erlotinib	phase 1b 42 NSCLC pts received T + E; 37 were c-MET+ (36 evaluable; 35 H-score ≥ 150, 1 MET amplified)	c-Met+ (central lab IHC H-score ≥150 or local lab MET amplification)		≥1	NR 95%CI: 2.8–NE	5.9 m 95CI: 1.2–NE	EGFR mut+: 34.5 (95%CI: 17.9–54.3) EGFR wildtype: 28.6% (95%CI: 3.7–71.0)
Wu et al. (2018). NCT01610336 [72]	Capmatinib (INC280) + Gefitinib	Phase Ib(61)/phaseII(100)(GCN < 4: *n* = 41 4 ≤ GCN < 6: *N* = 18; GCN ≥ 6: *n* = 36)	IHC, MET OE 2+ or 3+; FISH, MET Amp GCN ≥ 5, MET/(CEP7) ratio of ≥2:1 50% of tumor cells with IHC 3+ or MET GCN < 4)	EGFR-mutated advanced NSCLC	≥2 (≥1 prior EGFR-TKI)	Overall: 5.5 (95% CI, 3.8 to 5.6; mPFS in GCN ≥ 6 subgroup: 5.49 (95% CI, 4.21 to 7.29), mPFS in the 4 ≤ GCN < 6 subgroup: 5.39 (95% CI, 3.65 to 7.46); mPFS in the GCN < 4 subgroup: 3.91(95% CI, 3.65 to 5.55) mPFS in the IHC2+/GCN ≥ 5 subgroup: 7.29 (95%CI, 1.81 to 9.07) mPFS in the IHC3+ subgroup: 5.45 (95% CI, 3.71 to 7.10)	NR	Phase Ib/II overall: 43%; Phase II overall: 29%; GCN ≥ 6: 47%; 4 ≤ GCN < 6: 22%; GCN < 4: 12%; IHC 3+: 32%
Camidge et al. (2022) NCT01900652 [98]	emibetuzumab + erlotinib vs. emibetuzumab monotherapy	Phase II emibetuzumab + erlotinib (*n* = 83); emibetuzumab monotherapy (*n* = 28)	≥10% of cells expressing MET at ≥2+ by IHC	EGFR-mt NSCLC	≥1	3.3 vs. 1.6	NR	3.0 for emibetuzumab + erlotinib (95% CI: 0.4, 10.5) vs. 4.3% for emibetuzumab (95% CI: 0.1, 21.9)
**Combined therapies with third-generation EGFR-TKIs and MET-TKIs**								
Yu et al. (2021) ORCHARD Study [99]	Osimertinib + Savolitinib	phase II (*n* = 17)	NGS (criteria NR; GCN ranged from 7 to 68)	EGFRm	2 (progressed after prior first-line Osimertinib)	NR	NR	ORR: 41% (7/17)
Sequist et al. (2020) TATTON study NCT02143466 [71]	Osimertinib + Savolitinib	phase 1B; Part B *n* = 138 (osimertinib 80 mg psavolitinib 600 mg or 300 mg): (Part B1: previously received third generation EGFR TKI *n* = 69; part B2: no previous third-generation EGFR TKI, Thr790Met negative, *n* = 51; Part B3: no previous third-generation EGFR TKI, Thr790Met positive *n* = 18); Part D *n* = 42 (osimertinib plus savolitinib; no previous third-generation EGFR TKI, Thr790Met negative)	MET gene copy number gain ≥ 5 or MET/CEP7 ratio ≥ 2 by FISH; MET + 3 expression in ≥50% of tumor cells by IHC; ≥20% tumor cells, coverage of ≥200× sequencing depth and ≥5 copies of MET over tumor ploidy by NGS	EGFR mutation-positive(with or without T790M mutation)	≥2 (≥1 prior EGFR-TKI)	Part B overall: 5.5–11.1; Part D: 9.0 (95%CI: 5.4–12.9)	NR	part B: 33–67%; part D: 62%
E. Felip et al. (2019) NCT02335944 [100]	Capmatinib + Nazartinib *n* = 68 (66 had known MET status: 23 MET+, 43 METȡ)	Phase 1b/II study	MET+: IHC 3+ and/or GCN ≥4	EGFR-mutant stage IIIB/IV NSCLC	≥1	7.7 (95% CI: 5.4–12.2)	18.8 (95% CI:14.0–21.3)	43.5 (95% CI:23.2–65.5)
NCT03940703 INSIGHT 2 study [73]	Tepotinib plus Osimertinib vs. chemotherapy	Phase II (*n* = 425)	METamp by FISH testing (GCN ≥ 5 and/or MET/CEP7 ratio ≥ 2) or METamp determined by using NGS (GCN ≥ 2.3)	EGFR-mutated NSCLC	2	NR	NR	54.5% among the 22 patients with FISH detected MET amplification and at least 9 months of follow-up; 45.8% among the 48 participants with follow-up of 3 months or more; 50.0% for the 16 patients who were followed up for 9 months or more and 56.5% for the 23 followed up for 3 months or more
NCT03778229 SAVANNAH Study [101]	Osimertinib + Savolitinib	Phase II (*n* = 193)	High levels of MET overexpression and/or amplification, defined as IHC90+ and/or FISH10+, (IHC50+ and/or FISH5+; *n* = 193)	EGFRm+, MET+, progressed on prior Osimertinib	≥2	All patients (IHC50+ and/or FISH5+; *n* = 193): 5.3 (4.2, 5.8); Patients with high levels of MET (IHC90+ and/or FISH10+): (*n* = 108) 7.1 (5.3, 8.0) Patients with high levels of MET (IHC90+ and/or FISH10+) No prior chemo (*n* = 87): 7.2 (4.7, 9.2) Patients with lower levels of MET (*n* = 77): 2.8 (2.6, 4.3)	NR	Overall: All patients (IHC50+ and/or FISH5+; *n* = 193): 5.3 (4.2, 5.8) Patients with high levels of MET (IHC90+ and/or FISH10+) (*n* = 108): 7.1 (5.3, 8.0) Patients with high levels of MET (IHC90+ and/or FISH10+) No prior chemo (*n* = 87): 7.2 (4.7, 9.2) Patients with lower levels of MET (*n* = 77): 9 (4, 18)

Abbreviations: NSCLC, non-small cell lung cancer; ORR, objective response rate; mPFS, median progression-free survival; OS, median overall survival; MET, mesenchymal-epithelial transition factor; amp, amplification; EGFR, epidermal growth factor receptor; IHC, immunohistochemistry; FISH, fluorescence in situ hybridization; GCN, gene copy number; Pem, pemetrexed; Dox, docetaxel; Gem, gemcitabine; LBx, liquid biopsy; TBx, tissue biopsy; [CI], confidence interval; NR, not reported.

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
