# Peer review of "MET Amplification as a Resistance Driver to TKI Therapies in Lung Cancer: Clinical Challenges and Opportunities"

_cancers, 2023, doi:10.3390/cancers15030612_

Round 1

Reviewer 1 Report

The manuscript entitled " MET amplification as a resistance driver to TKI therapies in lung cancer: challenges and opportunities " by Dr. Xiuning Le. The authors aimed to clarify the underlying mechanisms of METamp-mediated resistance to tyrosine kinase inhibitors, discuss the ways and challenges in detection and diagnosis of MET amplifications in patients with metastatic NSCLC, summarize the recently published clinical data as well as the ongoing trials about new combination strategies overcome MET mediated TKIs resistance. In general, well written and logically presented review.

1. The purpose of this article is to clarify the biological mechanism of meth amplification mediated tyrosine kinase inhibitor resistance in NSCLC, but there is no discussion about the role of MET gene secondary amplification/copy number change in the formation of NSCLC targeted drug resistance. As the core content of this article, this part should be reviewed emphatically.

2. In this article, the author reviewed the drug combination strategies to overcome MET amplification mediated drug resistance, but did not discuss the relevant mechanisms of drug intervention.

3. The figure 1 could be enlarged for better visualization of the details due to its lack of sharpness.

4. The format of references is not uniform, please pay attention to modification.

Author Response

The manuscript entitled " MET amplification as a resistance driver to TKI therapies in lung cancer: challenges and opportunities " by Dr. Xiuning Le. The authors aimed to clarify the underlying mechanisms of METamp-mediated resistance to tyrosine kinase inhibitors, discuss the ways and challenges in detection and diagnosis of MET amplifications in patients with metastatic NSCLC, summarize the recently published clinical data as well as the ongoing trials about new combination strategies overcome MET mediated TKIs resistance. In general, well written and logically presented review.

  1. The purpose of this article is to clarify the biological mechanism of met amplification mediated tyrosine kinase inhibitor resistance in NSCLC, but there is no discussion about the role of MET gene secondary amplification/copy number change in the formation of NSCLC targeted drug resistance. As the core content of this article, this part should be reviewed emphatically.

Author response: In our manuscript, our current section “3. MET amplification as a mediator of resistance to targeted agents in NSCLC” was dedicated for the biology about the role of MET gene secondary amplification/copy number change in the formation of NSCLC targeted drug resistance. We acknowledge that this part is important to the review and expanded on the resistance mechanism, especially the mechanism via ERBB3 phosphorylation. We also expanded the section 6. Conclusion part to highlight that MET amplification can also mediate resistance to other targeted therapy.

  1. In this article, the author reviewed the drug combination strategies to overcome MET amplification-mediated drug resistance, but did not discuss the relevant mechanisms of drug intervention.

Author response: We added one paragraph to address the reviewer’s comment, elaborating on the mechanism of using dual inhibition to overcome MET amplification mediated targeted therapy resistance.

  1. The figure 1 could be enlarged for better visualization of the details due to its lack of sharpness.

Author response: we enlarged each component of figure 1 by about 50% per reviewer suggestion.

  1. The format of references is not uniform, please pay attention to modification.

Author response: Thank you for pointing this out. We aligned the references per reviewer suggestion.

Reviewer 2 Report

This is a well written review article focusing on the clinical relevance of MET gene amplification and mutations in lung cancer. The article is informative and novel of its kind, but highly focused on clinical applications. In consequence, there is an utter lack of discussion about any mechanism that may contribute to the observed phenotype.

Therefore, it would be my suggestion to highlight this fact already in the title. My suggestion would be to add the word "clinical":

MET amplification as a resistance driver to TKI therapies in lung cancer: CLINICAL challenges and opportunities

Author Response

This is a well written review article focusing on the clinical relevance of MET gene amplification and mutations in lung cancer. The article is informative and novel of its kind, but highly focused on clinical applications. In consequence, there is an utter lack of discussion about any mechanism that may contribute to the observed phenotype.

Therefore, it would be my suggestion to highlight this fact already in the title. My suggestion would be to add the word "clinical"

MET amplification as a resistance driver to TKI therapies in lung cancer: CLINICAL challenges and opportunities

Author response: we appreciate the reviewer’s input. We added the word ‘clinical’ in the title and named our review as ‘MET amplification as a resistance driver to TKI therapies in lung cancer: clinical challenges and opportunities’.

Reviewer 3 Report

Qin at al. give a comprehensive review focusing on Met amplification as mechanism of resistance to TKI inhibitors in NSCLC, however I have few comments: 

1) on paragraph 36: maybe "gents" would be "agents"

2) the format of Figure 1 legend should be adjust 

3) on section 3. "MET amplification as a mediator of resistance to targeted therapies in NSCLC" between paragraph 116-125 I would reinforce the concept of HER3-MET correlation. There is a particular study made by the group of Y.Yarden in 2020 showing that using anti-HER3 antibody in combination both in vitro and in vivo also downregulates MET expression. 

4) table 1 should be better organized 

Author Response

Qin at al. give a comprehensive review focusing on Met amplification as mechanism of resistance to TKI inhibitors in NSCLC, however I have few comments: 

  • on paragraph 36: maybe "gents" would be "agents"

Author response: thank you for pointing out this error. Edited.

  • the format of Figure 1 legend should be adjust 

Author response: we edited the legend by using bold font for the title of the legend.

  • on section 3. "MET amplification as a mediator of resistance to targeted therapies in NSCLC" between paragraph 116-125 I would reinforce the concept of HER3-MET correlation. There is a particular study made by the group of Y.Yarden in 2020 showing that using anti-HER3 antibody in combination both in vitro and in vivo also downregulates MET expression.

Author response: we added this line of evidence in the section 3.

  • table 1 should be better organized

Author response: we appreciate this input and separated the table to based on different generations of EGFR-TKIs and subtitled each part as “Combined therapies with first-generation EGFR-TKIs and MET-TKIs” and “Combined therapies with third-generation EGFR-TKIs and MET-TKIs”

Round 2

Reviewer 1 Report

Authors revised the manuscript, could accept for this

Author Response

Thank you. We sincerely appreciate all your valuable comments and suggestions which helped us in improving the quality of the manuscript.

Reviewer 3 Report

I do not have ant o there comments. The manuscript could be accept.

Author Response

(The authors gave the same response as above.)
